# Viscosity estimation model of fluorine-containing mold flux for continuous casting

**Zhongyu Zhao[1,2], Junxue Zhao[1]*, Boqiao Qu[1], Yaru Cui[1]**

**1** School of Metallurgical Engineering, Xi'an University of Architecture and Technology, Xi'an, Shaanxi, China, **2** CISDI Research & Development Co. Ltd., Chongqing, China

* zhaojunxue1962@126.com

**Data Availability Statement:** All relevant data are available within the paper and its Supporting Information files.

**Funding:** Thanks to National Natural Science Foundation of China (item No. 51674185,

## Abstract

A viscosity estimation model for fluorine-containing mold flux for continuous casting was investigated based on the Arrhenius formula and the rotating cylinder method combined with nonlinear regression analysis. This model is highly applicable and not limited by the slag of a certain composition. For most slag compositions, the viscosities estimated with this model deviated from the measured values by no more than 10%, which was in better agreement with the measured values than the viscosities estimated by the Riboud, Iida and Mills models. According to the model calculation and experimental detection, a viscosity isogram of $CaF_2$-$Na_2O$-$Al_2O_3$-$CaO$-$SiO_2$-$MgO$ slag was produced, and the mass fraction of $CaF_2$ in the low-viscosity zone was nearly 14%. An X-ray fluorescence spectrometric analysis of slag after the viscosity test showed that $CaF_2$ and $Na_2O$ were significantly reduced, and the measured viscosity was greater than the theoretical viscosity due to the volatilization.

## Introduction

The fluidity of slag has an important influence on the refining reaction, smelting temperature control, and heat and mass transfer of slag and metal during the steelmaking process [1–5]. Viscosity is the main factor affecting the fluidity of slag, and many scholars have analyzed the influence factors and control mechanisms of various kinds of slag viscosity with the objective of securing a universally applicable model to predict the viscosities of different kinds of slag changes under different temperatures [6–15]. The Riboud [16] proposed a viscosity model based on the Weymann Frenkel formula. Iida [17] put forward the viscosity model combined with crystallization temperature, and Mills [18] built a viscosity model based on optical basicity that widely used to estimate the viscosity of mold flux during continuous casting. However, each method has a certain limitation and scope of application, and there are significant differences between the viscosity estimations and the test results, especially for slag containing fluorine.

This article put forward a new viscosity model based on the Arrhenius equation and nonlinear regression analysis considering comprehensively the volatilization of fluorine-containing mold flux for continuous casting. An isoviscosity diagram is drawn according to this model and can provide theoretical support and practical basis for slag composition design and performance control.

51674186) for funding. The projects provided research fund and labor compensation for Zhao Zhongyu, Zhao Junxue, Qu Boqiao, and Cui Yaru but did not have any additional role in the study design, data collection and analysis, decision to publish, or preparation of the manuscript. The specific roles of these authors are articulated in the 'author contributions' section.

**Competing interests:** On behalf of all authors, I state that there is no conflict of interest. Zhao Zhongyu graduated from Xi'an University of Architecture and Technology, and now is working in CISDI Research & Development Co., Ltd. There are no any other relevant employment, consultancy, patents, products in development, or marketed products, etc. This does not alter our adherence to PLOS ONE policies on sharing data and materials.

## Materials and methods

The composition of fluorine-containing mold flux was designed in Table 1, according to the requirements and application of the continuous casting technique.

Twenty-seven viscosity tests (C1 ~ C27) were designed by quadratic regression orthogonal analysis [19] to obtain a viscosity estimation model based on the Arrhenius formula. To evaluate the effectiveness and applicability of this model, the viscosity data of fluorine-containing mold fluxes at the American National Physical Laboratory and the Department of Theoretical Metallurgy (M1 ~ M11) [20] were obtained and analyzed. At the same time, combined with the viscosity data of $CaF_2$-$CaO$-$SiO_2$ slag (S1–S7) [21], the uncertain relation of viscosity detection and slag volatilization was investigated. The compositions of the slags are given in Table 2.

The slags (C1 ~ C27) were prepared with chemical reagents according to Table 2, and the samples were ground by an agate ball mill at a speed of 200 r/min for 0.5 h, dried at 373 K for 5 h in GF101-2A Electric Blast Drying Oven, and then sealed and stored in the dark. An RTW-10 Melt Physical Property Comprehensive Analyzer was used for the viscosity experiment, and the heating rate was 10 K/min. It was cooled at 5 K/min when the temperature was reached to 1773 K, and the viscosity was recorded. High purity argon was used to protect the detection process, and the flow rate was set at 50ml / min.

## Results and discussion

### Viscosity analysis

Viscosity tests of the C1 ~ C27 slag samples were performed to obtain the linear relationship between the logarithm of viscosity (ln $\eta$) and the reciprocal of temperature ($1/T$) based on the Arrhenius equation (formulas 1 ~ 2). Taking the C1 slag viscosity test as an example, the results of the analysis are shown in Fig 1. The intercept (ln $A$) and slope ($B = E/R$) of each linear relationship were obtained in Table 3.

$$\eta = A\exp\left(\frac{E}{RT}\right) \tag{1}$$

$$\ln\eta = \ln A + \frac{E}{R} \times \frac{1}{T} \tag{2}$$

The viscosity parameters ln $A$ and $B$ were obtained by nonlinear regression analysis according to changes in the basicity (R) and composition, and the parameter models were as follows.

$$lnA = -6.69 + 0.07x_1 - 0.30x_2 - 0.93x_3 + 0.26x_4 - 0.08x_5 + 0.11x_1x_2 + 0.22x_1x_3 - 0.45x_1x_4 - 0.12x_1x_5$$
$$+0.05x_2x_4 - 0.01x_2x_5 - 0.02x_3x_4 + 0.01x_3x_5 + 0.01x_4x_5 - 0.05x_1^2 - 0.01x_2^2 + 0.04x_3^2 + 0.01x_5^2 \tag{3}$$

$$B = 17381.7 - 9875.3x_1 + 597.7x_2 + 1160.7x_3 - 456.8x_4 + 32.0x_5 - 229.1x_1x_2 - 234.3x_1x_3 + 758.5x_1x_4 + 367.4x_1x_5$$
$$+2.59x_2x_3 - 82.6x_2x_4 + 15.7x_2x_5 + 30.93x_3x_4 - 10.7x_3x_5 - 28.6x_4x_5 + 2772.7x_1^2 + 15.34x_2^2 - 52.1x_3^2 - 5.4x_4^2 - 22.1x_5^2 \tag{4}$$

$x_1$, $x_2$, $x_3$, $x_4$, and $x_5$ represent the basicity R and the mass fractions of $Al_2O_3$, $CaF_2$, $Na_2O$, and MgO, respectively. The above regression equations were analyzed with SPSS statistical software, for P = 0.0001 < 0.05, and correlation coefficient R = 0.9941, which showed a good fit. Fig 2 was drawn to clarify the influence of each component on the parameters $lnA$ and $B$.

**Table 1. Designed composition of fluorine-containing mold flux.**

| Component | $CaF_2$ | R($CaO/SiO_2$) | $Na_2O$ | $Al_2O_3$ | MgO |
|---|---|---|---|---|---|
| Mass fraction/% | 4~20 | 0.6~1.2 | 3~12 | 2~12 | 0~12 |

**Table 2. Components of mold fluxes for continuous casting, wt %.**

| Slag | CaO | SiO$_2$ | Al$_2$O$_3$ | CaF$_2$ | Na$_2$O | MgO | (R) | Slag | CaO | SiO$_2$ | Al$_2$O$_3$ | CaF$_2$ | Na$_2$O | MgO | (R) |
|------|-----|---------|-------------|---------|---------|-----|-----|------|-----|---------|-------------|---------|---------|-----|-----|
| C1 | 24.6 | 22.3 | 11.2 | 19 | 11.7 | 11.2 | 1.1 | C24 | 33.7 | 37.4 | 7.2 | 12.4 | 3.1 | 6.2 | 0.9 |
| C2 | 34.0 | 30.9 | 10.5 | 17.8 | 4.7 | 2.1 | 1.1 | C25 | 27.6 | 30.7 | 7.6 | 13 | 8.1 | 13 | 0.9 |
| C3 | 35.1 | 32.0 | 11.2 | 7.8 | 11.7 | 2.2 | 1.1 | C26 | 33.8 | 37.5 | 7.6 | 13 | 8.1 | 0 | 0.9 |
| C4 | 35.1 | 31.9 | 10.5 | 7.3 | 4.7 | 10.5 | 1.1 | C27 | 30.7 | 34.1 | 7.6 | 13 | 8.1 | 6.5 | 0.9 |
| C5 | 32.8 | 29.8 | 4.5 | 19 | 11.7 | 2.2 | 1.1 | M1 | 23.4 | 38.3 | 6.2 | 12.9 | 18.8 | 0.5 | 0.6 |
| C6 | 32.9 | 29.9 | 4.2 | 17.8 | 4.7 | 10.5 | 1.1 | M2 | 17.5 | 40.0 | 5.2 | 14.5 | 21.5 | 1.3 | 0.4 |
| C7 | 33.9 | 30.9 | 4.5 | 7.8 | 11.7 | 11.2 | 1.1 | M3 | 20.1 | 34.4 | 4.8 | 14.3 | 26.4 | 0.0 | 0.6 |
| C8 | 42.8 | 38.9 | 4.2 | 7.3 | 4.7 | 2.1 | 1.1 | M4 | 19.4 | 36.6 | 18.0 | 17.2 | 8.8 | 0.0 | 0.5 |
| C9 | 23.0 | 32.9 | 11.2 | 19 | 11.7 | 2.2 | 0.7 | M5 | 32.6 | 31.2 | 5.3 | 8.0 | 22.9 | 0.0 | 1.0 |
| C10 | 23.3 | 33.2 | 10.5 | 17.8 | 4.7 | 10.5 | 0.7 | M6 | 22.5 | 42.2 | 10.4 | 10.7 | 13.0 | 1.3 | 0.5 |
| C11 | 23.9 | 34.2 | 11.2 | 7.8 | 11.7 | 11.2 | 0.7 | M7 | 21.5 | 33.2 | 3.7 | 15.4 | 25.7 | 0.5 | 0.6 |
| C12 | 31.0 | 44.4 | 10.5 | 7.3 | 4.7 | 2.1 | 0.7 | M8 | 22.7 | 38.6 | 6.3 | 13.5 | 18.9 | 0.0 | 0.6 |
| C13 | 22.1 | 31.5 | 4.5 | 19 | 11.7 | 11.2 | 0.7 | M9 | 22.3 | 35.0 | 8.9 | 16.5 | 17.3 | 0.0 | 0.6 |
| C14 | 29.3 | 41.9 | 4.2 | 17.8 | 4.7 | 2.1 | 0.7 | M10 | 20.4 | 34.0 | 4.3 | 17.9 | 23.4 | 0.0 | 0.6 |
| C15 | 30.4 | 43.4 | 4.5 | 7.8 | 11.7 | 2.2 | 0.7 | M11 | 13.8 | 32.0 | 3.5 | 20.2 | 27.1 | 3.5 | 0.4 |
| C16 | 30.2 | 43.1 | 4.2 | 7.3 | 4.7 | 10.5 | 0.7 | S1 | 33.5 | 53.0 | - | 13.5 | - | - | 0.6 |
| C17 | 35.3 | 29.5 | 7.6 | 13 | 8.1 | 6.5 | 1.2 | S2 | 32.9 | 47.8 | - | 19.3 | - | - | 0.7 |
| C18 | 24.3 | 40.5 | 7.6 | 13 | 8.1 | 6.5 | 0.6 | S3 | 33.1 | 46.9 | - | 20.0 | - | - | 0.7 |
| C19 | 28.1 | 31.3 | 13 | 13 | 8.1 | 6.5 | 0.9 | S4 | 30.8 | 37.8 | - | 31.4 | - | - | 0.8 |
| C20 | 33.3 | 36.9 | 2.2 | 13 | 8.1 | 6.5 | 0.9 | S5 | 27.0 | 30.4 | - | 42.6 | - | - | 0.9 |
| C21 | 26.6 | 29.6 | 7.6 | 21.6 | 8.1 | 6.5 | 0.9 | S6 | 33.0 | 39.6 | - | 27.4 | - | - | 0.8 |
| C22 | 34.8 | 38.7 | 7.6 | 4.3 | 8.1 | 6.5 | 0.9 | S7 | 33.2 | 38.6 | - | 28.2 | - | - | 0.9 |
| C23 | 27.5 | 30.5 | 8 | 13.6 | 13.6 | 6.8 | 0.9 | | | | | | | | |

(Note: the wt% Na$_2$O of samples M1 to M11 corresponds to the sum of wt% Na$_2$O and wt% K$_2$O)

It can be seen from Fig 2A that *lnA* is between -10 and -12, and only five samples have relatively larger deviation, namely C17 [alkalinity]$_{max}$, C18 [alkalinity]$_{min}$, C21 [CaF$_2$(wt%)]$_{max}$, C24 [Na$_2$O(wt%)]$_{min}$, and C26 [MgO(wt%)]$_{min}$. In particular, C21 corresponds to the largest *lnA*, C24 corresponds to the minimum *lnA*. The fitting points fluctuation in Fig 2B is opposite to that in Fig 2A. Therefore, CaF$_2$ and Na$_2$O are two significant factors effecting the viscosity of mold flux.

## Comparison of viscosity estimation models

Combined with the viscosity tests (C1 ~ C27) and the viscosity data of fluorine-containing mold fluxes (M1 ~ M11), this viscosity estimation model can be evaluated and compared with the traditional viscosity models. The results are shown in Fig 3 below.

$$\Delta = \frac{1}{N} \sum_{n=1}^{N} \frac{(\eta_n)_{est} - (\eta_n)_{mea}}{(\eta_n)_{mea}} \tag{5}$$

In comparing the above models to estimate the viscosity of different types of fluorine-containing continuous casting mold fluxes, the deviation rate of the Riboud model was relatively larger. The estimated viscosity of the Iida model was higher than the measured viscosity, and the estimated value of the Mills model was lower than the measured value. In contrast, the deviation rate between the estimated and measured viscosity was less than 10%, and this model could better describe the viscosity change of different fluorine-containing continuous casting mold fluxes.

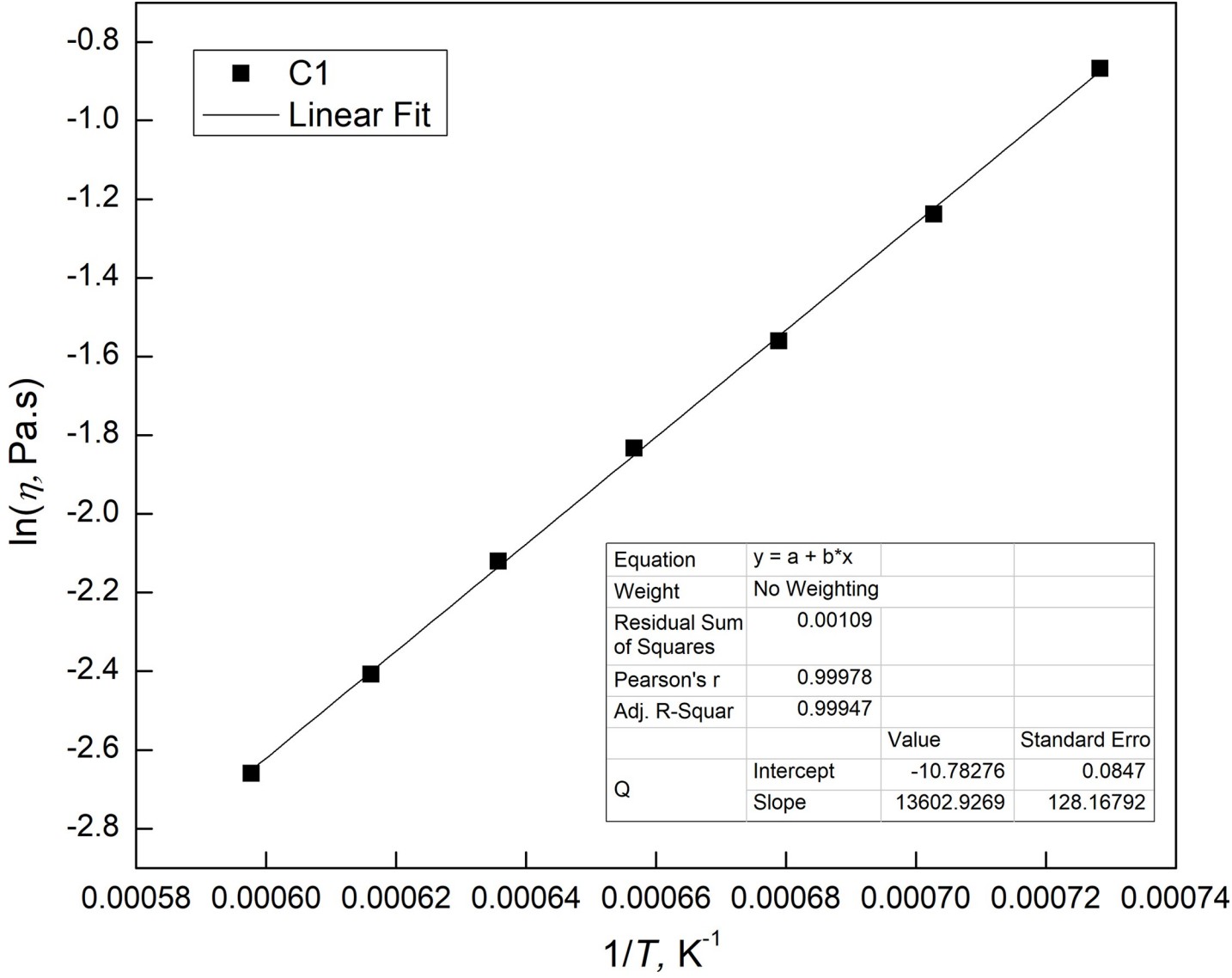

**Fig 1. Linear analysis of lnη to 1/T.**

**Table 3. Viscosity controlling parameters.**

| Slag | ln$A$ | $B$ | Slag | ln$A$ | $B$ | Slag | ln$A$ | $B$ |
|------|-------|-----|------|-------|-----|------|-------|-----|
| C1 | -10.7828 | 13602.93 | C10 | -11.1663 | 15523.76 | C19 | -11.6419 | 16899.2 |
| C2 | -11.1521 | 15331.25 | C11 | -11.4069 | 16399.51 | C20 | -10.5736 | 14696.67 |
| C3 | -11.4827 | 16323.8 | C12 | -11.9045 | 19019.28 | C21 | -9.26063 | 11473.14 |
| C4 | -11.2338 | 16106.31 | C13 | -10.7114 | 13670.59 | C22 | -11.3388 | 18048.45 |
| C5 | -10.9666 | 13904.19 | C14 | -11.1896 | 15932.56 | C23 | -10.4814 | 13697.65 |
| C6 | -11.1806 | 14859.89 | C15 | -11.4391 | 17000.89 | C24 | -17.2362 | 25991.79 |
| C7 | -11.3522 | 15328.48 | C16 | -11.1312 | 16491.46 | C25 | -10.5812 | 14320.87 |
| C8 | -11.3635 | 16421.96 | C17 | -14.961 | 21103.44 | C26 | -13.0682 | 18316.6 |
| C9 | -10.9992 | 14722.03 | C18 | -12.8972 | 19711.71 | C27 | -12.1166 | 16750.76 |

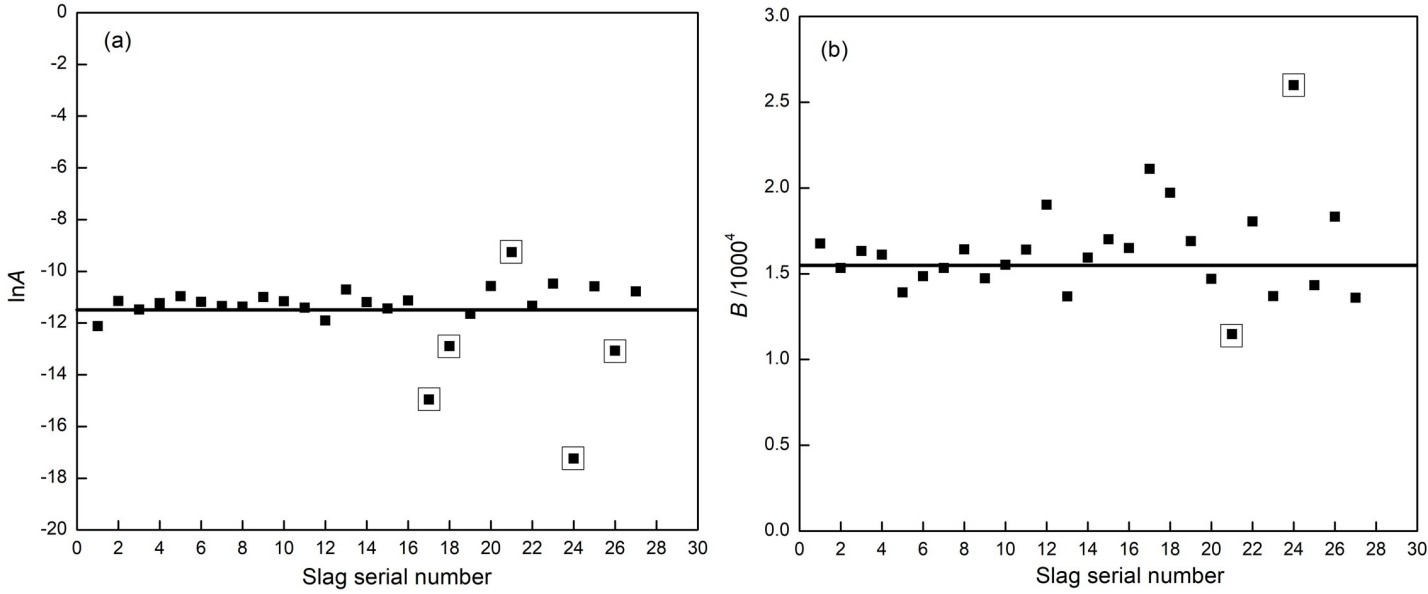

**Fig 2. Viscosity parameter fitting value.** (a) *lnA*; (b) *B*.

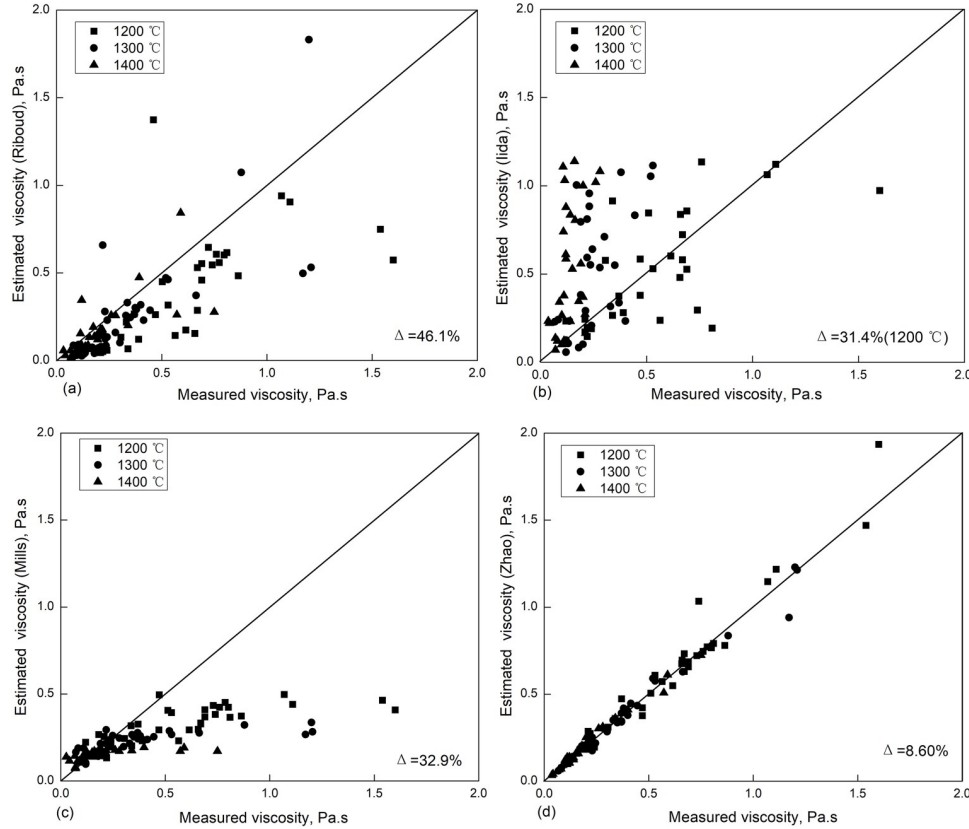

**Fig 3. Estimated and measured viscosity of fluorine-containing mold flux.** (a) Riboud model; (b) Iida model; (c) Mills model; (d) studied model. (the Δ in the figures above represents the deviation rate between the estimated viscosity and the measured viscosity, and the calculation formula (5) is shown below).

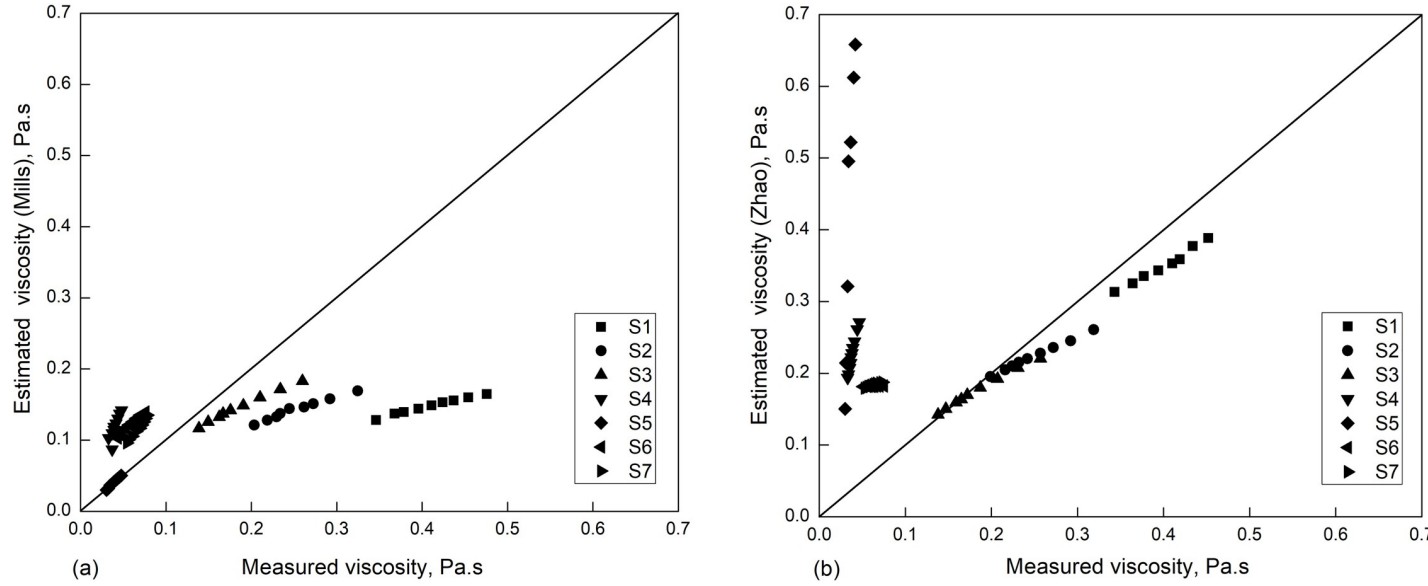

**Fig 4. Estimated and measured viscosity of CaF$_2$- CaO- SiO$_2$ slag.** (a) Mills model; (b) studied model.

At the same time, the viscosity data of CaF$_2$-CaO-SiO$_2$ slag (S1~S7) were compared and analyzed by the Mills model and Zhao model, respectively, as shown in Fig 4.

It can be seen that the Mills model had a better fitting on the viscosity of high fluorine content slag (S5), and the viscosity fitting by the Zhao model was better for medium and low fluorine content slag (S1 ~ S3). This was mainly affected by the volatility of the fluorine-containing slag; the higher the fluorine content is, the stronger the slag volatilization and the greater the change in composition, eventually leading to a deviation between the estimated and measured viscosity.

## Application of this viscosity estimation model

The above viscosity estimation model can determine the influence of different component changes on the viscosity of the mold flux based on C27 slag with a fixed basicity R = 0.9 and MgO (wt%) = 6.5, and the CaF$_2$, Na$_2$O and Al$_2$O$_3$ components impact the slag viscosity in Fig 5.

It can be seen from the first part of Fig 5C that the increase in CaF$_2$ can significantly reduce the viscosity of the mold flux. However, the influence of Al$_2$O$_3$ and Na$_2$O on viscosity was restricted by the CaF$_2$ content in the slag system. In Fig 5A, the CaF$_2$ content is 17% at the intersection point. On the left side of the intersection, i.e. CaF$_2$(wt%)<17%, the increase in Al$_2$O$_3$ significantly increased the viscosity of the slag. If the CaF$_2$ content was more than 17%, the viscosity decreased with the addition of Al$_2$O$_3$. In Fig 5B, the CaF$_2$ content is 11.5% at the intersection point. The viscosity of the slag system decreased significantly with increasing Na$_2$O mass when the CaF$_2$ content was more than 11.5%, and if the CaF$_2$ content was less than 11.5%, the effect of Na$_2$O on viscosity was insignificant.

According to this viscosity estimation model and the experimental data, an isoviscosity diagram of CaF$_2$-Na$_2$O-Al$_2$O$_3$-CaO-SiO$_2$-MgO at 1773 K can be drawn in Fig 6, similarly, with a fixed basicity R = 0.9 and MgO (wt%) = 6.5.

It can be seen from the isoviscosity curves in Fig 6 that the mass fraction of CaF$_2$ in the low viscosity zone was nearly 14%, and a level of CaF$_2$ that is too high or too low will increase the viscosity of the slag system. At the same time, the viscosity will increase gradually near the Al$_2$O$_3$ angle. The viscosity of the mold flux can be reduced by rationally adjusting the components according to the above viscosity isogram.

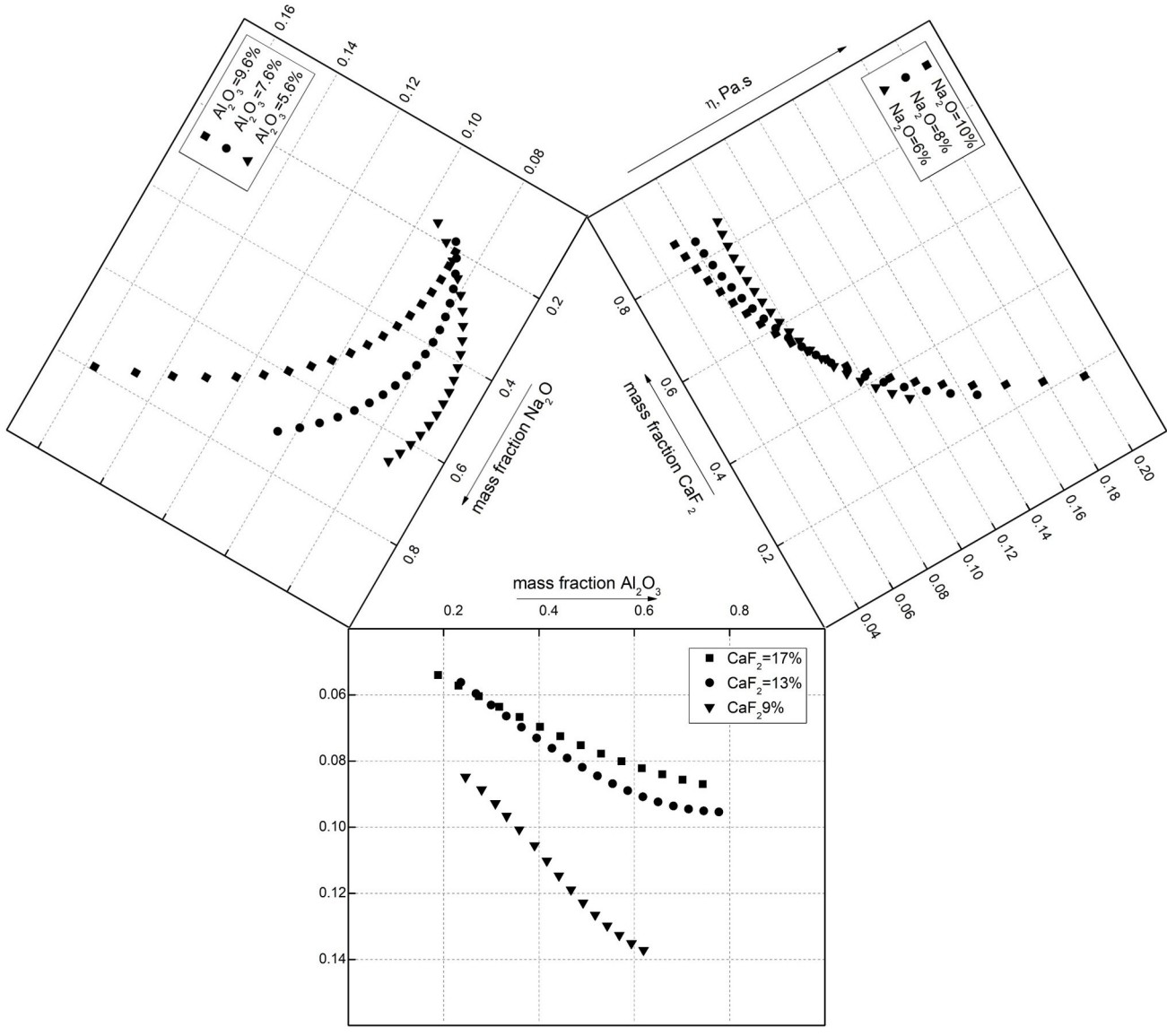

**Fig 5. Effect of CaF₂, Na₂O and Al₂O₃ on the viscosity of mold flux at 1773 K.**

However, the volatilization leads to the change of slag composition, resulting in the measured viscosity cannot correspond to the initial composition of slag, as shown in Fig 4. It is necessary to detect the composition of samples after viscosity tests.

## Viscosity model modification based on volatilization

The composition of samples after viscosity tests was carried out by XRF analysis due to the volatilization of fluoride in the viscosity measurements, and the results are shown in Table 4.

The viscosity model (1) ~ (2) can be modified according to the data in the table above, as shown below.

$$lnA = -7.05 - 1.33x_1 - 1.08x_2 - 0.47x_3 + 0.43x_4 - 0.25x_5 - 0.35x_1x_2 - 0.32x_1x_3 + 0.56x_1x_4 - 0.2x_1x_5$$
$$-0.01x_2x_3 + 0.02x_2x_4 - 0.01x_3x_5 + 0.02x_4x_5 + 2.69x_1{}^2 + 0.09x_2{}^2 + 0.06x_3{}^2 - 0.16x_4{}^2 + 0.04x_5{}^2 \tag{6}$$

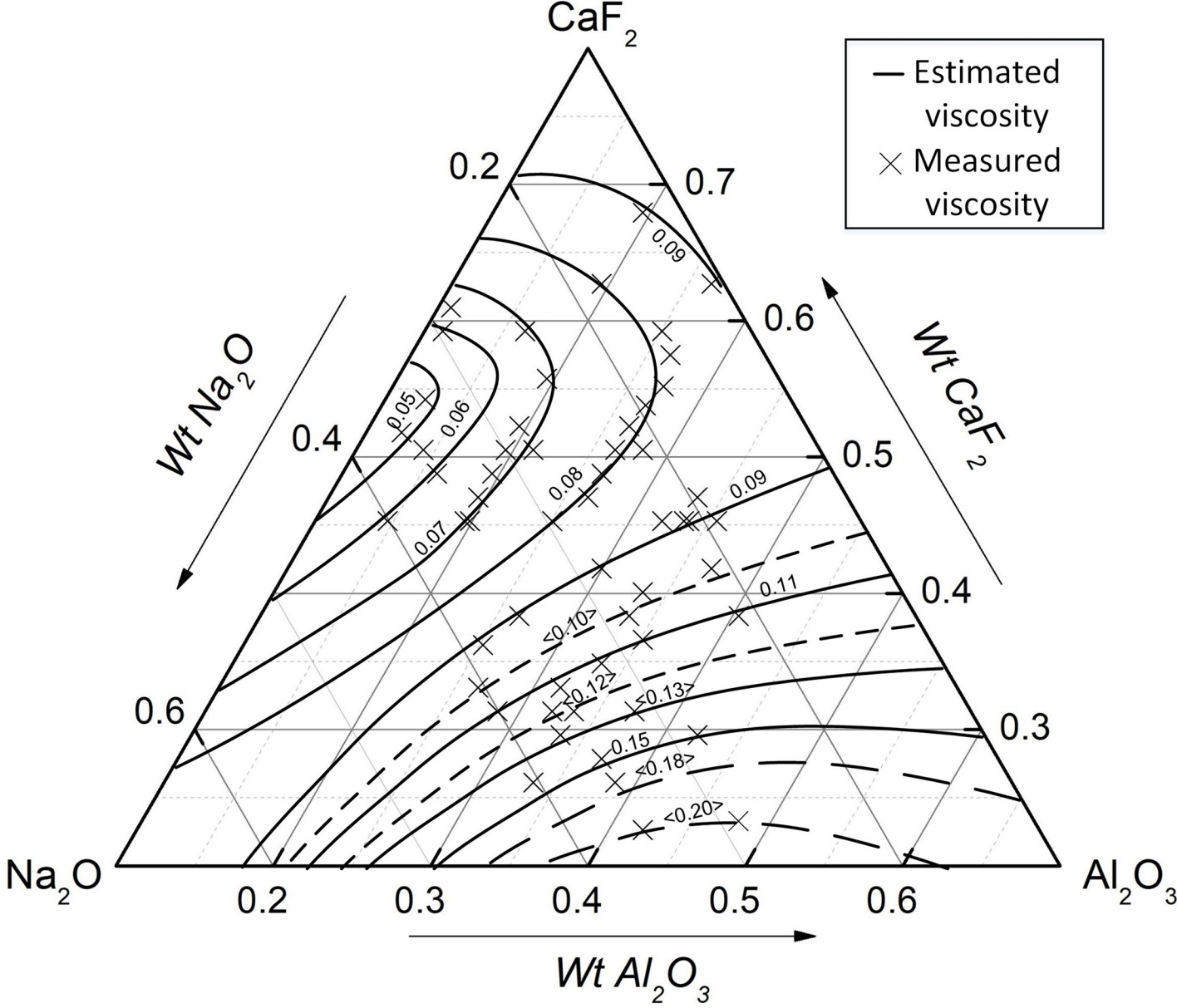

**Fig 6. CaF₂-Na₂O-Al₂O₃-CaO-SiO₂-MgO isoviscosity diagram (R = 0.9, MgO(wt%) = 6.5, 1773 K).**

$$B = 11257.7 + 3200.1x_1 + 1942.2x_2 + 429.5x_3 - 1038.1x_4 + 405.8x_5 + 477.4x_1x_2 + 696.9x_1x_3 - 786.5x_1x_4 + 351.1x_1x_5$$
$$+16.9x_2x_3 - 42.1x_2x_4 - 4.4x_2x_5 + 0.9x_3x_4 + 22.7x_3x_5 - 20.3x_4x_5 - 6651.4x_1^2 - 141.4x_2^2 - 99.0x_3^2 + 262.5x_4^2 - 76.9x_5^2 \tag{7}$$

Similarly, the isoviscosity diagram of mold flux at 1773 K can be drawn in Fig 7 based on the above model with a fixed basicity R = 0.9 and MgO (wt%) = 6.5.

Fig 7 shows the viscosity corresponding to the actual composition of the mold flux, and Fig 6 shows the viscosity corresponding to the initial composition. Comparing the isoviscosity curves in Figs 6 and 7, it can be seen that Na₂O decreased significantly during the viscosity

**Table 4. Components of mold fluxes after viscosity tests, wt %.**

| Slag | CaO | SiO₂ | Al₂O₃ | CaF₂ | Na₂O | MgO | (R) | Slag | CaO | SiO₂ | Al₂O₃ | CaF₂ | Na₂O | MgO | (R) |
|------|-----|------|-------|------|------|-----|-----|------|-----|------|-------|------|------|-----|-----|
| C1 | 29.9 | 23.6 | 11.9 | 14.7 | 8.1 | 11.9 | 1.3 | C15 | 38.9 | 47.2 | 4.9 | 0.4 | 6.3 | 2.4 | 0.8 |
| C2 | 38.6 | 32.3 | 11.0 | 14.3 | 1.5 | 2.2 | 1.2 | C16 | 36.5 | 45.9 | 4.5 | 1.7 | 0.2 | 11.2 | 0.8 |
| C3 | 41.1 | 33.8 | 11.8 | 2.9 | 8.1 | 2.3 | 1.2 | C17 | 46.7 | 32.7 | 8.4 | 4.1 | 0.8 | 7.2 | 1.4 |
| C4 | 41.8 | 34.0 | 11.2 | 1.7 | 0.1 | 11.2 | 1.2 | C18 | 32.2 | 44.0 | 8.3 | 6.0 | 2.4 | 7.1 | 0.7 |
| C5 | 40.4 | 32.1 | 4.8 | 13.4 | 7.0 | 2.4 | 1.3 | C19 | 36.2 | 33.9 | 14.1 | 6.2 | 2.5 | 7.1 | 1.1 |
| C6 | 38.3 | 31.5 | 4.4 | 13.7 | 1.0 | 11.1 | 1.2 | C20 | 41.3 | 39.9 | 2.4 | 6.6 | 2.8 | 7.0 | 1.0 |
| C7 | 39.2 | 32.5 | 4.7 | 3.4 | 8.5 | 11.8 | 1.2 | C21 | 34.3 | 32.0 | 8.2 | 15.7 | 2.7 | 7.0 | 1.1 |
| C8 | 50.0 | 41.5 | 4.5 | 1.7 | 0.1 | 2.2 | 1.2 | C22 | 39.6 | 40.5 | 8.0 | 0.1 | 5.0 | 6.8 | 1.0 |
| C9 | 29.0 | 35.1 | 11.9 | 14.1 | 7.6 | 2.3 | 0.8 | C23 | 42.1 | 35.3 | 9.2 | 1.3 | 4.2 | 7.9 | 1.2 |
| C10 | 28.4 | 35.2 | 11.1 | 13.5 | 0.7 | 11.1 | 0.8 | C24 | 37.9 | 39.0 | 7.5 | 9.0 | 0.1 | 6.5 | 1.0 |
| C11 | 28.5 | 35.9 | 11.8 | 3.5 | 8.6 | 11.8 | 0.8 | C25 | 32.4 | 32.3 | 8.0 | 8.9 | 4.7 | 13.7 | 1.0 |
| C12 | 36.3 | 46.7 | 11.1 | 2.7 | 1.0 | 2.2 | 0.8 | C26 | 39.4 | 39.6 | 8.0 | 8.5 | 4.4 | 0.0 | 1.0 |
| C13 | 26.0 | 32.9 | 4.7 | 15.8 | 9.0 | 11.7 | 0.8 | C27 | 37.0 | 36.3 | 8.1 | 7.8 | 3.8 | 6.9 | 1.0 |
| C14 | 35.5 | 44.6 | 4.5 | 12.9 | 0.2 | 2.2 | 0.8 | | | | | | | | |

test. Moreover, Tables 2 and 4 show that CaF₂ also decreased. Therefore, it can be seen that the measured viscosity in Fig 6 was relatively larger than the theoretical viscosity in Fig 7 due to volatilization.

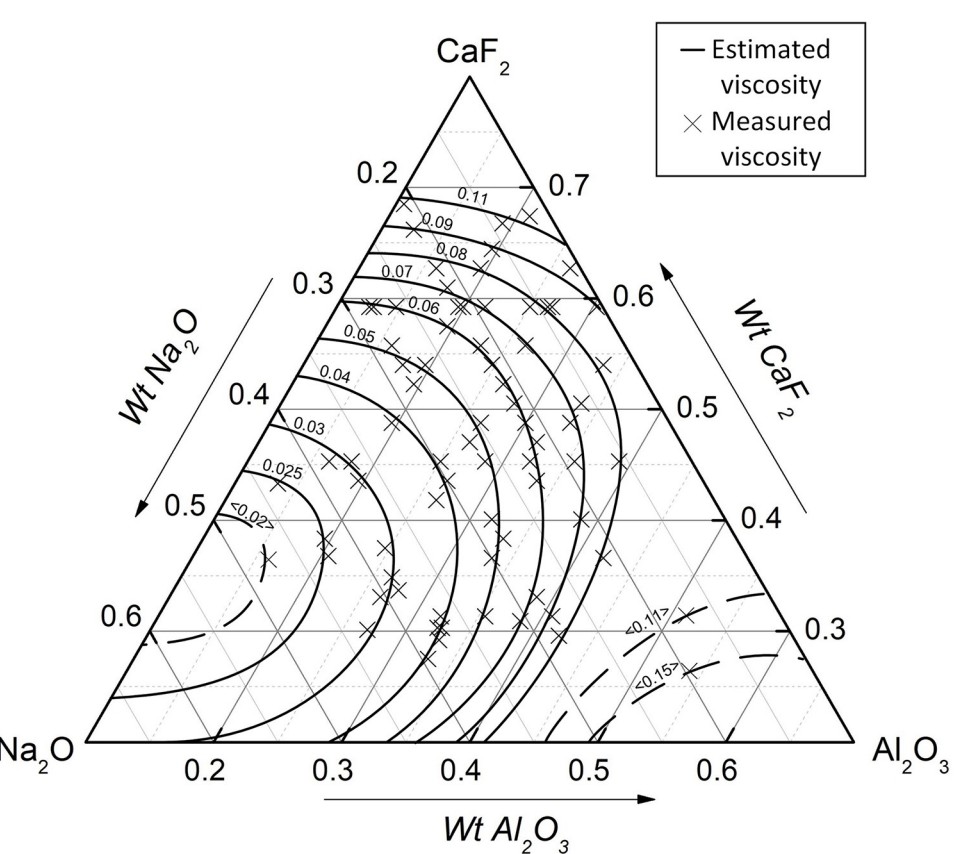

**Fig 7. Modified isoviscosity diagram (R = 0.9, MgO(wt%) = 6.5, 1773 K).**

## Conclusions and prospects

1. A viscosity estimation model of fluorine-containing mold flux for continuous casting was investigated through viscosity detection and nonlinear regression analysis based on the Arrhenius equation. The viscosities estimated with this model were within 10% of the measured values, which achieves better agreement with the measured values than the viscosities estimated by the Riboud, Iida and Mills models.

2. $CaF_2$ can significantly reduce the viscosity of mold flux, and the influence of $Al_2O_3$ and $Na_2O$ on viscosity was restricted by the $CaF_2$ content in the slag system. Moreover, the $CaF_2$-$Na_2O$-$Al_2O_3$-$CaO$-$SiO_2$-$MgO$ isoviscosity diagram at 1773 K was drawn through this model estimation and experimental detection, and the $CaF_2$ mass fraction was close to 14% in the low viscosity zone.

3. The viscosity model and isoviscosity diagram were modified according to the XRF analysis of slag after the viscosity test. After contrast and analysis, it was found that $Na_2O$ and $CaF_2$ decreased significantly, and the measured viscosity was larger than the theoretical viscosity due to mold flux volatilization.

4. The effect of volatilization on slag viscosity is significant. Therefore, it is necessary to conduct further research on the volatilization mechanism to control the physicochemical properties of slag and obtain a more accurate viscosity estimation model.

## Supporting information

**S1 File.**
(RAR)

## Author Contributions

**Investigation:** Boqiao Qu.

**Project administration:** Junxue Zhao.

**Writing – original draft:** Zhongyu Zhao.

**Writing – review & editing:** Yaru Cui.

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
