## [Decision Letter · Decision Letter 0]

11 Dec 2020

PONE-D-20-28340

Viscosity Detection and Estimation Model of Fluorine-containing Mold Flux for Continuous Casting

PLOS ONE

Dear Dr. Zhao,

Thank you for submitting your manuscript to PLOS ONE. After careful consideration, we feel that it has merit but does not fully meet PLOS ONE’s publication criteria as it currently stands. Therefore, we invite you to submit a revised version of the manuscript that addresses the points raised during the review process.

As pointed out by the reviewers, please, make an effort to clarify the novelty of this work. In its current state, the novelty of the work is unclear.

We look forward to receiving your revised manuscript.

Kind regards,

Antonio Riveiro Rodríguez, PhD

Academic Editor

PLOS ONE

Journal Requirements:

2.) Thank you for stating the following in the Acknowledgments Section of your manuscript:

'Thanks to National Natural Science Foundation of China (item No. 51674185, 51674186) for funding

and the Key Discipline of Metallurgical Engineering in Shaanxi Province.'

'NO - The funders had no role in study design, data collection and analysis, decision to publish, or preparation of the manuscript.'

3.) In your Data Availability statement, you have not specified where the minimal data set underlying the results described in your manuscript can be found. PLOS defines a study's minimal data set as the underlying data used to reach the conclusions drawn in the manuscript and any additional data required to replicate the reported study findings in their entirety. All PLOS journals require that the minimal data set be made fully available. For more information about our data policy, please see http://journals.plos.org/plosone/s/data-availability.

4.) PLOS requires an ORCID iD for the corresponding author in Editorial Manager on papers submitted after December 6th, 2016. Please ensure that you have an ORCID iD and that it is validated in Editorial Manager. To do this, go to ‘Update my Information’ (in the upper left-hand corner of the main menu), and click on the Fetch/Validate link next to the ORCID field. This will take you to the ORCID site and allow you to create a new iD or authenticate a pre-existing iD in Editorial Manager. Please see the following video for instructions on linking an ORCID iD to your Editorial Manager account: https://www.youtube.com/watch?v=_xcclfuvtxQ

5.) Thank you for stating the following in the Competing Interests section:

'NO authors have competing interests'

We note that one or more of the authors are employed by a commercial company: CISDI Thermal & Environmental Engineering Co. Ltd

Reviewers' comments:

Reviewer's Responses to Questions

**Comments to the Author**

1. Is the manuscript technically sound, and do the data support the conclusions?

Reviewer #1: Partly

Reviewer #2: Partly

2. Has the statistical analysis been performed appropriately and rigorously? 

Reviewer #1: Yes

Reviewer #2: I Don't Know

3. Have the authors made all data underlying the findings in their manuscript fully available?

Reviewer #1: Yes

Reviewer #2: No

4. Is the manuscript presented in an intelligible fashion and written in standard English?

Reviewer #1: Yes

Reviewer #2: Yes

5. Review Comments to the Author

Reviewer #1: Title: Title well reflects the study.

Abstract

: Please shorten the abstract (avoid redundant words).

: Avoid abbreviations in the abstract/conclusions.

Readers first glance these sections and may confuse with the unknown abbreviations.

: Avoid redundant sentences/words. It may annoy the readers and consume time. (readers') and space (for the journal).

: Use of wrong words/unclear sentences

English/Grammar/spelling/scientific communication style.

: Authors may benefit by checking the grammar/spelling using a computer programme. e.g. Grammarly, ProWritingAis, etc.

: Avoid beginning a sentence (especially a paragraph) with conjoining words. e.g. For, As, When, etc.

Figures

: Captions should have stand-alone meaning

: Quality should be improved—unclear axis titles and poor image quality.

The reader will confuse what are neta and 1/T please define them in the title. Quality should be improved.

Tables

: Captions should have stand-alone meaning. e.g. Did the authors mean composition?

: Quality should be improved.

Writing style

: Do not repeat the heading in the text of the body.

: Unclear sentences.

: Redundant words/sentences. e.g."as shown in Table 1" instead please write just " Table 1"

: Too many equations.

: Please leave a space between the number and scientific units, including for % sign. More information: https://physics.nist.gov/cuu/Units/checklist.html

: The clear and concise writing style is required.

: Please try to shorten the text by combining sentences, without losing the meaning. Remove any redundant words or sentences.

Originality:

: Please make it clear if there is any connection to this article: https://pubs.acs.org/doi/10.1021/acs.energyfuels.0c00926

: : The paragraph just after Fig.4 is somewhat matching with this: https://link.springer.com/book/10.1007%2F978-3-319-48769-4

: An unusually high percentage of text matching with other sources, although some can be ignored, for example, experimental procedures in the authors' previous article but some needs to be corrected.

Literature survey

: Should be strengthened (very short) and no new references, e.g., 2020

Conclusions

: Good

: There is more than one conclusion; "Conclusion"? It should be "Conclusions".

: Please shorten the conclusion/s. Maybe in bullet points form.

: Clear, concise and explicit jargon should be used.

Recommendation: The article needs major revision before considering for publication. The article is too short.

Reviewer #2: The topic is of interest to the metallurgical community especially for those who work in the determination of physicochemical properties of slags. However, major modifications must be made to the manuscript in its current form.

In the Experimental part, the temperature range in which the viscosity was determined is not mentioned. It is assumed, according to Fig.1 that it is between 1100-1400ºC. But experimental data at 1773 K (1500ºC) are reported in Fig. 5 (from where were the experimental data in Fig. 5 taken?).

It should be clarified that the compositions of samples M1 to M11 (Table 2) the wt% Na2O corresponds to the sum of wt% Na2O and wt% K2O of the original work (Reference 20).

The X-axis of Fig.1 should be 1/T, but it is labeled ln (1/T). (Should be corrected).

In the Y axis of Fig.1 it is necessary to mention the viscosity units (Pa.s?). Please, complete.

In the same Fig. 1 the parameters: lnA (= Intercept = -10.7828) and B (= Slope = 13602.93) correspond to sample C1 (according to Table 3), and not to sample C27 as it is mentioned in the text.

The expressions for the calculation of lnA and B should be revised. They do not coincide with those analyzed in the tables, according to the chemical composition.

The section: "Application of this viscosity estimation model", should be discussed more extensively and in greater depth.

In line 108, it is mentioned: "When the CaF2 content was more than 17%, the viscosity decreased with the addition of Al2O3". However this effect is not shown in Fig. 4. where, for CaF2 <17 wt% the viscosity always increases with increasing Al2O3 addition.

It would have to provide more details on the calculation methodology to obtain Fig.5 and Fig.6.

6. PLOS authors have the option to publish the peer review history of their article (what does this mean?). If published, this will include your full peer review and any attached files.

Reviewer #1: No

Reviewer #2: No

---

## [Author Response · Author response to Decision Letter 0]

9 Jan 2021

Thanks to the two reviewers for their comments and suggestions and we have modified this article and answered all the questions one by one in the file named Response to Referees.

---

## [Decision Letter · Decision Letter 1]

1 Feb 2021

PONE-D-20-28340R1

Viscosity Detection and Estimation Model of Fluorine-containing Mold Flux for Continuous Casting

PLOS ONE

Dear Dr. Zhao,

Thank you for submitting your manuscript to PLOS ONE. After careful consideration, we feel that it has merit but does not fully meet PLOS ONE’s publication criteria as it currently stands. Therefore, we invite you to submit a revised version of the manuscript that addresses the points raised during the review process.

As commented by one of the reviewers, I also agree that the word detection should be removed from the paper title and from the title of section  "Viscosity detection and analysis". Viscosity is not detected, it is measured. On the other hand, the introduction should  be improved. A proper review of the state of the art should be discussed in the introduction to highlight the novelty of the work and the new approach taken by this work. In the current state, this section is not suitable.

We look forward to receiving your revised manuscript.

Kind regards,

Antonio Riveiro Rodríguez, PhD

Academic Editor

PLOS ONE

Reviewers' comments:

Reviewer's Responses to Questions

**Comments to the Author**

1. If the authors have adequately addressed your comments raised in a previous round of review and you feel that this manuscript is now acceptable for publication, you may indicate that here to bypass the “Comments to the Author” section, enter your conflict of interest statement in the “Confidential to Editor” section, and submit your "Accept" recommendation.

Reviewer #1: (No Response)

Reviewer #2: All comments have been addressed

2. Is the manuscript technically sound, and do the data support the conclusions?

Reviewer #1: No

Reviewer #2: (No Response)

3. Has the statistical analysis been performed appropriately and rigorously? 

Reviewer #1: No

Reviewer #2: (No Response)

4. Have the authors made all data underlying the findings in their manuscript fully available?

Reviewer #1: Yes

Reviewer #2: (No Response)

5. Is the manuscript presented in an intelligible fashion and written in standard English?

Reviewer #1: No

Reviewer #2: (No Response)

6. Review Comments to the Author

Reviewer #1: The word "Detection" on the title does not seem to be appropriate; may be deleted.

Scientific communication style should be improved; very colloquial jargon. Clarity of presentation should be improved.

Previous comments have not yet been appropriately addressed. Literature study should be strengthened.

The study is weak to derive conclusions.

Reviewer #2: The authors have correctly followed the suggestions made on the first version of the manuscript.

I understand that the current version is ready to be published

7. PLOS authors have the option to publish the peer review history of their article (what does this mean?). If published, this will include your full peer review and any attached files.

Reviewer #1: No

Reviewer #2: **Yes: **EDGARDO BENAVIDEZ

---

## [Author Response · Author response to Decision Letter 1]

2 Feb 2021

1. We accept the proposal that the word detection has been removed from the paper title and from the title of section “Viscosity detection and analysis”.

2. We have modified the introduction to summarize and compare the previous research, and highlight the novelty and application of the work.

---

## [Decision Letter · Decision Letter 2]

15 Feb 2021

Viscosity Estimation Model of Fluorine-containing Mold Flux for Continuous Casting

PONE-D-20-28340R2

Dear Dr. Zhao,

We’re pleased to inform you that your manuscript has been judged scientifically suitable for publication and will be formally accepted for publication once it meets all outstanding technical requirements.

Kind regards,

Antonio Riveiro Rodríguez, PhD

Academic Editor

PLOS ONE

Reviewers' comments:

Reviewer's Responses to Questions

**Comments to the Author**

1. If the authors have adequately addressed your comments raised in a previous round of review and you feel that this manuscript is now acceptable for publication, you may indicate that here to bypass the “Comments to the Author” section, enter your conflict of interest statement in the “Confidential to Editor” section, and submit your "Accept" recommendation.

Reviewer #1: All comments have been addressed

2. Is the manuscript technically sound, and do the data support the conclusions?

Reviewer #1: Partly

3. Has the statistical analysis been performed appropriately and rigorously? 

Reviewer #1: Yes

4. Have the authors made all data underlying the findings in their manuscript fully available?

Reviewer #1: Yes

5. Is the manuscript presented in an intelligible fashion and written in standard English?

Reviewer #1: Yes

6. Review Comments to the Author

Reviewer #1: The manuscript has improved from its first version. The analysis was carried out well. Could have improved the literature a bit better; the number of references seems too low for a journal paper.

7. PLOS authors have the option to publish the peer review history of their article (what does this mean?). If published, this will include your full peer review and any attached files.

Reviewer #1: No

---

## [Editor Report · Acceptance letter]

18 Feb 2021

PONE-D-20-28340R2 

Viscosity Estimation Model of Fluorine-containing Mold Flux for Continuous Casting 

Dear Dr. Zhao:

I'm pleased to inform you that your manuscript has been deemed suitable for publication in PLOS ONE. Congratulations! Your manuscript is now with our production department. 

Kind regards, 

on behalf of

Dr. Antonio Riveiro Rodríguez 

Academic Editor

PLOS ONE